# Cold therapy and pain relief after hot-iron disbudding in dairy calves

**Kane P. J. Colston** [1], **Thomas Ede** [2], **Michael T. Mendl** [1], **Benjamin Lecorps** [1]*

**1** Bristol Veterinary School, University of Bristol, Bristol, United Kingdom, **2** Department of Clinical Studies, Swine Teaching and Research Center, School of Veterinary Medicine, University of Pennsylvania, Kennett Square, PA, United States of America

* b.lecorps@bristol.ac.uk

## Abstract

Even when pain control is implemented, calves may experience pain for days after hot-iron disbudding. Whether calves seek pain relief post-disbudding offers a novel approach to assessing pain in these animals. By employing an approach-aversion paradigm, we explored the ability of cold therapy to provide immediate pain relief in disbudded calves. Calves were habituated to the manual placement of a cool or ambient pack on their forehead for a short duration simultaneous to milk reward consumption, prior to disbudding. Calves were then disbudded under local anaesthesia (procaine) and analgesia (meloxicam), and responses to the packs were observed over subsequent days. Individual calves were consistently exposed to either cool or ambient packs in different halves of a two-sided experimental pen, allowing for the testing of approach-aversion and conditioned place preference. We found calves approached milk rewards quicker and maintained contact for longer when receiving cold therapy compared to the ambient control. However, calves did not display any conditioned preference for the pen where they received the cool pack. These results add to the growing evidence of lasting pain following disbudding procedures and suggests that cold therapy provides some form of pain relief post-disbudding. Future studies should seek other ways to use cold therapy post-disbudding to reduce aversiveness and human involvement.

## Introduction

Hot-iron disbudding is a routine husbandry procedure in young cattle, involving the removal of horn buds and thermal destruction of underlying tissue to prevent horn growth [1]. Disbudding is conducted primarily as a safety precaution, preventing potential injury to other cattle or people [2]. Nevertheless, disbudding has been shown to have negative consequences on calf welfare, impacting long-term neurophysiology [3, 4], social behaviours [5, 6] and affective state [7–9]. Whilst administration of local anaesthetic (cornual nerve block) and nonsteroidal anti-inflammatory drug (NSAID) analgesia may alleviate pain during and post-disbudding [10], the efficacy of such measures are not guaranteed [11] and long-term pain as well as hyperalgesia are regularly reported [3, 12–14]. Yet, use of pain mitigation remains low, with only 14% of farmers and 56% of veterinarians in the UK using NSAIDs for disbudding, with

electronic supplementary information and are available to reviewers and to anyone at publication.

**Funding:** This research was funded by the Animal Welfare Research Network's Kick-start funding award and the University of Bristol through the start-up research funds attributed to Benjamin Lecorps.

**Competing interests:** The authors have declared that no competing interests exist.

cost and workload cited as primary factors in a farmer's decision to forgo using pain relief [15–17].

Characterising pain in animals is difficult, given that they cannot report what they feel and may not display visible signs of pain in the presence of humans [18]. Physiological changes such as plasma cortisol concentration [19], heart rate variability [4], wound temperature [20], and behavioural observations such as ear flicking, head-shaking and kicking [13, 21, 22] are typically relied upon instead to infer the presence of pain [23]. However, both spontaneous behaviours and physiological evidence provide weak inferences for the emotional component of pain in animals [24].

More recently, interest has developed in the underlying emotional states of calves following disbudding. Studies have showed that calves experience low mood and anhedonia hours to days after disbudding [8, 25, 26]. Conditioned place preference (CPP) tests–a commonly used paradigm to assess what animals remember from past experiences [27]–have also indicated that post-procedural pain and its associated emotional impact are remembered by calves [7]. Using the CPP paradigm, calves have also shown preference for the provision of analgesia with disbudding [28], as well as prolonged pain control (i.e., in the form of a local anaesthetic injection) up to 20 days after disbudding when compared to sham disbudded controls [14], suggesting that they were still in pain.

Cold therapy–typically running water between 2–15°C–is regularly used in humans and animals to suppress inflammation and associated pain following burn injuries [29, 30]. Whilst less regularly used in animals for analgesic purposes, cold therapy is suggested to be "an effective analgesic tool for acute pain management" [31], and appears suitable to cautery disbudding given the nature of the injury associated with the procedure. With running water being likely impractical for the disbudded wound [32], cold spray as a pre-disbudding pain mitigation strategy has been trialled previously in calves with indications of analgesic properties [33].

The aim of our study was to investigate whether cold therapy, through the manual application of a cool pack to the disbudding wound, would be able to provide instantaneous pain relief on the days following the procedure. For this we conducted an experimental control trial, using a methodology combining an approach-aversion test and a conditioned place preference procedure. Calves were offered a milk reward which they could access at the 'price' of having a gel pack (at cold or ambient temperature) in contact with their wound site. As previously noted, calves display increased wound sensitivity after disbudding [34]. By presenting calves a conflicting choice between events of positive (milk reward) and negative (contact pain) valences, the difference in motivational trade-offs between the two pack temperatures could inform us on whether cold therapy provides pain relief. Our assumption was that the inherent value of the cool and ambient pack would change after disbudding from mildly aversive to aversive for the ambient pack, and from mildly aversive to rewarding (because of pain relief) for the cool pack. We expected that, following disbudding, calves would approach the milk reward associated with application of the cool pack quicker and spend more time in contact with the cool pack, in comparison with the aversive ambient pack. In the CPP test, we expected calves to spend a greater proportion of time in the coloured pen associated with the cool pack.

## Materials and methods

### Ethics statement

The study was approved by The University of Bristol AWERB Committee (# UIN/22/072) and calves were cared for according to the standards of the University's dairy farm. All disbudding took place as part of routine management at the farm.

## Animals and housing

This study was conducted from April to July 2023 at the University of Bristol's Wyndhurst Farm in Langford, UK. No previous studies have used exactly the same paradigm for this topic, making *a priori* power calculations difficult. Using the *pwr* package in R [33], a minimum sample size of 17 individuals was calculated for a statistical power of 0.8, a small effect size (f2 = 0.02) and a significance level of 0.05 using linear models for our two main variables (latency and post-drinking contact duration in the approach-aversion test). Due to farm constraints, we could enrol a total of 15 calves (11 Holstein, 3 British blue, 1 Longhorn, birthweight of (mean ± SD) 40.7 ± 5.64kg) at (mean ± SD)) 58.1 ± 15.5 d of age. One animal was excluded from the study post-disbudding because she completely stopped approaching the milk feeder buckets during sessions post-disbudding.

Experiments were conducted over the duration of six days (Monday to Saturday). Calves were given *ad libitum* access to water, grain, and straw, as well as a total of 7.5L of milk replacer per day during the experimental sessions (Sprint Plus 50, Bridgmans Farm Direct; 150gL⁻¹), split equally between experimental sessions in the morning (0800 h) and afternoon (1600 h). This was identical to the feeding program delivered to calves prior to experimentation. Milk replacer was mixed with warm water (45°C) as stated by the provider. Experiments were conducted at mealtimes (8am and 4pm), so when calves did not receive their full milk allowance (3.75L per meal) during an experimental session, they were fed the remaining amount just after the session ended. Calves were housed in straw bedded hutches in groups of 2–4 (hutch size 2.1 x 2.4m) with access to an outside courtyard (3m x 2m). Fresh straw was added to the pens throughout the week.

## Experimental set-up

Experimental sessions were conducted on each calf individually within the test area consisting of three adjoining sections, separated by wooden doors (Fig 1). These sections were comprised of a concrete-floored lobby and two colour-coded straw-bedded test pens ("Blue" and "White"). A milk feeder bucket was attached at the far end of each test pen, visible from the lobby area when the adjoining doors were opened fully.

Cold therapy was applied through physical contact between the calf's head and a gel pack (The Heat Pack Company, UK; from herein referred to as 'pack', SM1). The pack was maintained at a constant temperature of between 4–5°C for the duration of the session ('Cool pack') as well as an ambient control pack at approximately 20°C ('Ambient pack'). Cool pack temperatures were monitored using an infrared thermometer (SOVARCATE HS960D), confirming that pack temperatures did not change by more than 1°C during a given session.

## Experimental procedure

For each calf, a total of eleven sessions took place across six days (Fig 2). Calves were tested individually and returned to their home pen at the end of every session. Each session consisted of six trials per calf (for a total of 66 trials per calf). Each calf was *a priori* pseudorandomly allocated a colour/treatment combination and an order of treatment, ensuring a balanced experimental design. At the beginning of the first trial the calf waited in the lobby area (Fig 1). The access door to a test pen was opened, granting access to the bucket containing 0.5L of milk replacer, with the adjoining door between the two test pens closed. After entry by the calf, the access door was closed. For the duration of the trial, when contact was made between the calf and the nipple of the milk bucket, the pack was manually placed on top of the calf's horn buds with light pressure (see S1 Video for illustrative videos). When contact between the calf and the nipple was lost, the pack was removed. When the milk bucket was empty, calves were given a further minute to keep suckling or wait in the test pen ('post-drinking' time). During

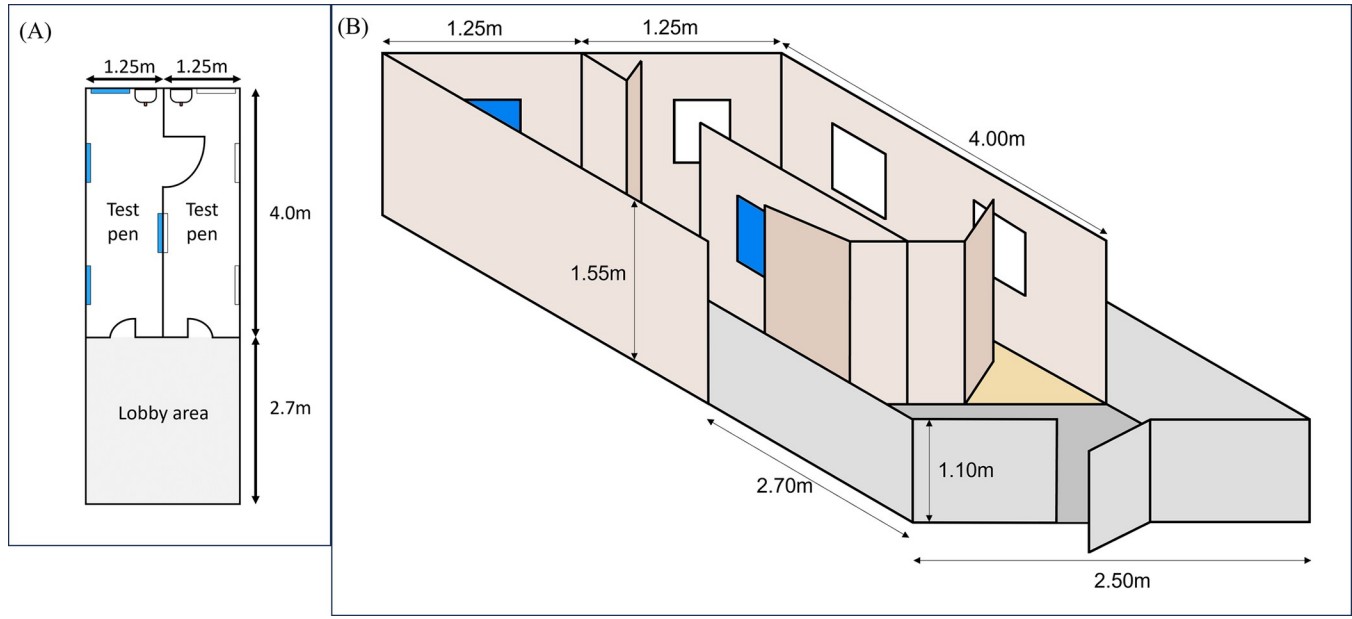

**Fig 1. Experimental enclosure.** (A) aerial diagram of test pen and adjoining lobby area with positioning of doors and milk buckets (to scale); (B) three-dimensional projection of test pens and lobby area (not to scale). The coloured boards were fixed to the pen walls, positioned for calves to better distinguish the left and right side of the pen from memory. The colours are therefore a proxy for left vs right side of the experimental pen.

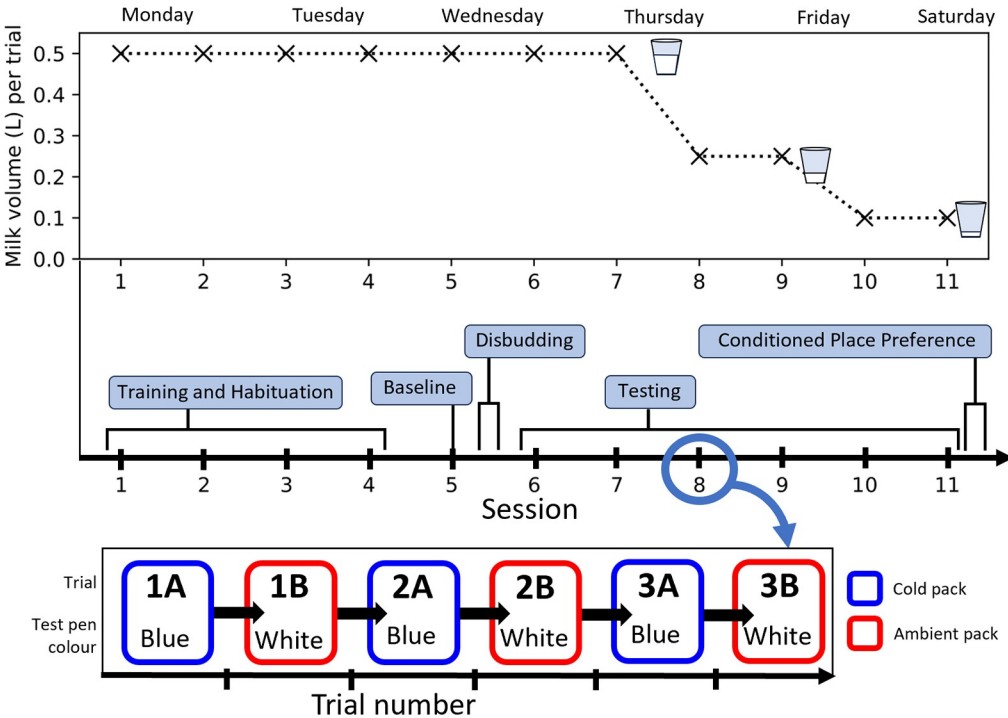

**Fig 2. Timeline illustrating the experimental procedure.** The top graph illustrates the volume of milk given to calves per trial in each experimental session. Numbers on the x-axis correspond to each experimental session. Session 1–4 represent the training and habituation phase. Session 5 corresponds to the pre-disbudding baseline session. Disbudding occurs between session 5 and 6. Session 6–11 correspond to the testing phase. The conditioned place preference test occurred immediately following session 11. Each session took approximately 20 minutes per calf. The layout of each session is also illustrated, with each session consisting of 6 separate trials, alternating between a cool pack (blue box) and ambient (red box). In this example, the cold pack was associated with the blue test pen and the ambient pack associated with the white test pen. These pen colour/treatment paired associations, as well as the pack temperature of the first trial, were pseudorandomised and allocated evenly across participants.

this time, the pack was applied to the calf's head whenever they were in contact with the nipple. At the end of the minute, the access door to the lobby area was reopened and the calf exited the test pen. This was repeated with calves on alternating sides of the test area for a total of six trials. They thus experienced three trials in one pen with the cool pack and three trials in the other pen with the ambient pack, alternating between pens from trial to trial.

### Training/habituation phase

The first four experimental sessions (Monday morning; Monday afternoon; Tuesday morning; Tuesday afternoon) were assigned for habituation and training. Calves were habituated to the experimental procedure and fed milk replacer using the feeder buckets inside the test pen. Cold and ambient packs were introduced during feeding from Monday afternoon–in line with prior pseudorandomised allocation–to habituate the calves to physical contact and to allow them to begin forming associations between colour and treatment.

### Baseline and disbudding

The session following the training/habituation phase (Session 5, Wednesday morning, Fig 2) was used as the pre-disbudding baseline session. Calves were given 0.5L per trial with 3 trials in each chamber as previously described. Hot-iron disbudding was performed using a hand-held gas iron (Portasol Gas Dehorner, Ritchey) immediately following session 5 by a trained veterinarian, with horn buds scooped out. Calves were administered a subcutaneous meloxi-cam analgesic (Metacam, 0.5 mgkg$^{-1}$) and a procaine cornual nerve block (Procamidor Duo,160 mg per horn bud), five minutes prior to disbudding.

### Test phase

The test phase ran for the remaining six sessions (Wednesday afternoon to Saturday morning). For sessions 6–7, calves received 0.5L of milk replacer per trial. For sessions 8–9, this was reduced to 0.25L per trial, then reduced further to 0.1L per trial for sessions 10–11 (Fig 2), to investigate motivation to suckle (and thus make contact with the pack) with diminishing reward, as has been conducted in previous approach-aversion tasks [35].

Following session 11, each calf was placed in the lobby area and allowed to choose which side of the test pen to enter. Both doors to the lobby area were then closed, securing the calf in the test pen, with the adjoining door between the two sides of the pen clamped open. The calf was given free access to the test pens for 15 minutes, testing for conditioned place preference (CPP) using which pen they chose to enter first and the proportion of time spent in each pen.

### Data collection

Latency to drink (herein, 'latency') was recorded using a stopwatch from the moment the calf in the lobby turned its head and directed its gaze towards the open door, until suckling commenced. We took the *a priori* decision that, if a calf took more than 30s to enter the pen and contact the bottle, this was interpreted as a 'no-go' response, ending the trial. The calf would then be returned to the lobby area to commence the next trial, consistent with previous studies [26]. Latencies cut off in this way were included as 30s ceiling values for subsequent statistical analysis (15/594 trials, 7/15 of which were ambient trials).

A GoPro HERO9 camera (GoPro, San Mateo, CA) was positioned directly above each milk feeding bucket for the duration of each session (S1 Video). The drinking duration (sL$^{-1}$) was extracted from video using the time taken to drink the milk replacer and the volume provided. If the calf stopped drinking before the milk bucket was empty, calves were given one minute to

reinitiate drinking, otherwise the trial ended, with a 60 second ceiling value recorded (27/594 trials, 14/27 of which were ambient trials). The consumption of 0.5L of milk within 60s is well within previously reported calf drinking rates [36, 37] and was therefore deemed an appropriate cut-off duration. The number of contacts with the nipple ("nipple contacts") was determined in the subsequent minute after the bucket was emptied, based on the number of times the calf placed its mouth on the nipple. During the minute following the emptying of the bucket, the duration with which the calf continued to suckle and therefore experienced contact with the pack was extracted from the video data ("post-drinking contact duration"). The number of nipple contacts or post-drinking contact duration were not recorded for trials which ended due to latency exceeding 30s or pauses in milk reward consumption exceeding 60s.

During the CPP test, the same camera was positioned above the test pens. Over the 15-minute duration, the number of seconds spent within each side of the pen was extracted from the video. Calves were considered to be within a given side if both front feet crossed the threshold. We also recorded the colour of pen that calves initially chose to enter from the lobby area. Calves were fed at least 10 minutes before being tested for conditioned place preference. This was done to prevent calves from seeking food in the arena during the test.

## Statistical analysis

We first tested the effect of pack treatment ('Cool' vs 'Ambient') and colour ('Blue' vs 'White') on the latency, drinking duration, nipple contacts and post-drinking contact duration during the baseline (session 5, prior to disbudding) using a mixed model approach. Through comparison of AIC/BIC criteria, we selected a linear mixed model (LMM) for post-drinking contact duration, whilst a generalised linear mixed model (GLMM) was used for the other three outcome variables. To model latency and drinking duration, we selected a gamma distribution to account for positive skewness. Nipple contacts was modelled using a poisson distribution. In all cases, we included 'Calf' and 'Trial' as random effects. These pre-disbudding tests were conducted to establish if a baseline aversion or motivation for the test treatment was present, or if any underlying experimental bias existed between the two sides of the test pen.

Next, we investigated the behavioural change in response to the two treatments (cool or ambient pack) from pre- to post-disbudding. For each of the outcomes (latency, drinking duration, post-drinking contact duration and nipple contacts), a baseline average was independently calculated for each calf across the three trials within session 5. To avoid assumptions of statistical equivalence between the two treatments, each calf had two baseline values (an 'Ambient' baseline and a 'Cool' baseline). The absolute change was calculated for each of sessions 6 to 11 using the baseline average values from session 5, with each calf and treatment calculated separately. Next, the data was transformed through the addition of a constant to eliminate negative and null values (to fit the gamma distribution). Mixed models were then constructed for each outcome variable, which included milk volume per trial, colour ('Blue' vs 'White'), treatment ('Cool' vs 'Ambient') and the 2-way interaction between milk volume per trial and treatment as fixed effects. Both 'Calf' and the nested effect of 'Trial' within 'Session' were included in each model as random effects. AIC/BIC criteria were used to determine the type of model selected (LMM or GLMM). Similarly to the pre-disbudding models, a linear mixed model was selected for post-drinking contact duration, whilst latency, nipple contacts and drinking duration were best modelled using a Gamma-distribution GLMM. To account for discrepancy between the calves' perceived volume of milk upon entry to the test pen and the true volume after it was decreased in session 8 and 10, we included 'expected milk volume' as a fixed effect for the latency model, rather than milk volume. The influence of both 'Calf' and 'Trial' nested within 'Session' as random effects were tested using likelihood ratio tests. In

all models, both random effects significantly improved model fit and so remained within the final model structure. Construction of LMMs and GLMMs was performed using 'lme4' package in R [38]. Quantile-quantile (Q-Q) plots and residuals were inspected to establish normality and homoscedasticity where appropriate. For latency, a second post-disbudding GLMM was also constructed, which excluded the 30 second ceiling values due to concerns about their influence on model fit and interpretability. This model better fit the latency data with regards to AIC/BIC criteria and Q-Q plots (see S1 Table).

To explore further whether differences between the three trials within each treatment may help to explain any observed differences between treatments, we performed a graphical analysis by plotting both treatments ('Cool' vs. 'Ambient') for each trial separately for any test metrics that were indicated to have a significant influence in the linear models.

To test for CPP, a Shapiro-Wilk test was first conducted to test the assumption of normality. A one-sample t-test was then performed to establish whether the time spent in the cool compartment was significantly different from 50%. A Chi-squared test on calf entry choice was conducted to test whether calves preferentially entered the test pen associated with cool pack treatment.

Because experimenters could not be blinded to pack temperature, we conducted inter-rater reliability tests for post-drinking contact duration, number of contacts and drinking duration. Latency data could not be tested because it was recorded live. 30 randomly selected trials across all calves were used and coded by an observer blind to treatment. We calculated the intraclass correlation coefficient (ICC) between the blinded and original values for the 30 randomly selected trials for each measure. We found a strong inter-rater reliability for drinking duration (ICC = 0.913), post-drinking contact duration (ICC = 0.932) and number of contacts (ICC = 0.871).

## Results

### Pre-disbudding

Prior to disbudding, treatment pack had no influence on latency (t = -1.352, P = 0.187), post-drinking contact duration (t = -1.745, P = 0.086) or nipple contacts (t = -0.117, P = 0.907), but did influence drinking duration (estimate ± SE = 0.09 ± 0.05 s, t = 2.048, P = 0.041), suggesting calves took longer to finish drinking the milk reward with the cool pack prior to disbudding. Colour of the pen did affect latency (t = 2.021, P = 0.043)—with calves approaching the blue pen faster on average, but did not impact post-drinking contact duration (t = 0.896, P = 0.374), nipple contacts (t = 0.904, P = 0.366) or drinking duration (t = -0.337, P = 0.736).

### Post-disbudding

Six trials were excluded following data collection due to technical issues, resulting in a total of 588 trials across all individuals. Pack temperature influenced the change in latency (t = -3.22, 95% CI: -0.321, -0.078, P = 0.001) and the change in post-drinking contact duration following disbudding (t = 3.193, 95% CI: 1.797, 7.507, P = 0.002), indicating that calves went faster and chose to stay longer with the cool pack post-disbudding (Fig 3). The influence of pack temperature on changes in latency was still observed when 30 second ceiling values were excluded from the analysis (t = -2.11, 95% CI: -0.181, -0.007, P = 0.035; S1 Table). No effects of treatment were found for drinking duration (P = 0.562) and nipple contacts (P = 0.292). Decreasing milk volume was associated with decreases in drinking duration (t = -4.411, P < 0.001), post-drinking contact duration (t = 2.977, P = 0.024) and the number of nipple contacts (t = 4.096, P = 0.004), but the expected milk volume had no detectable effects on latency (P = 0.918). Furthermore, no significant interaction effect between milk volume and treatment was observed

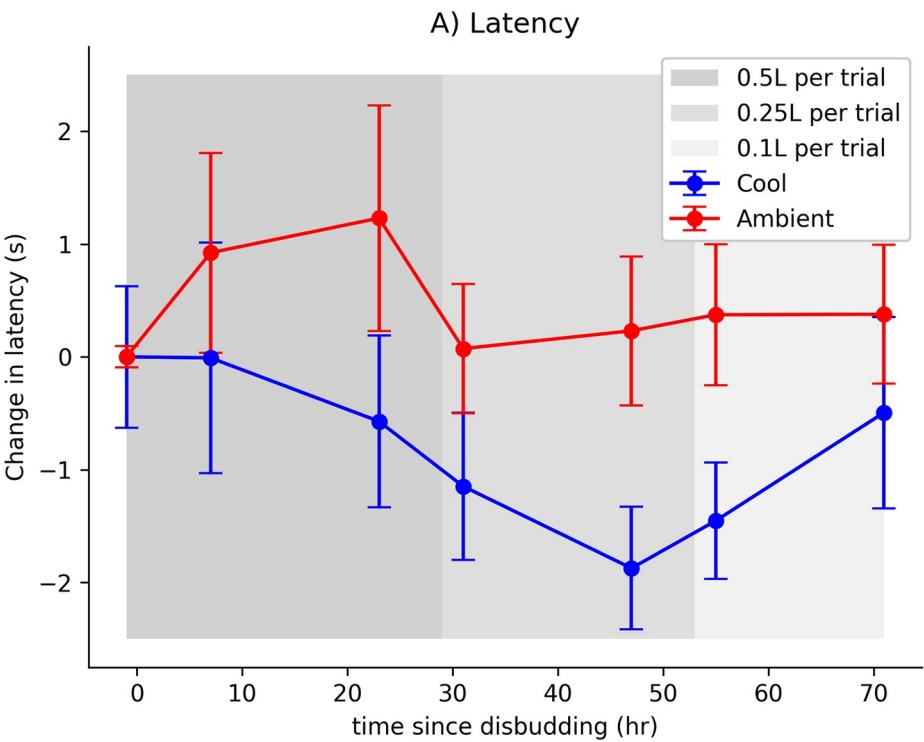

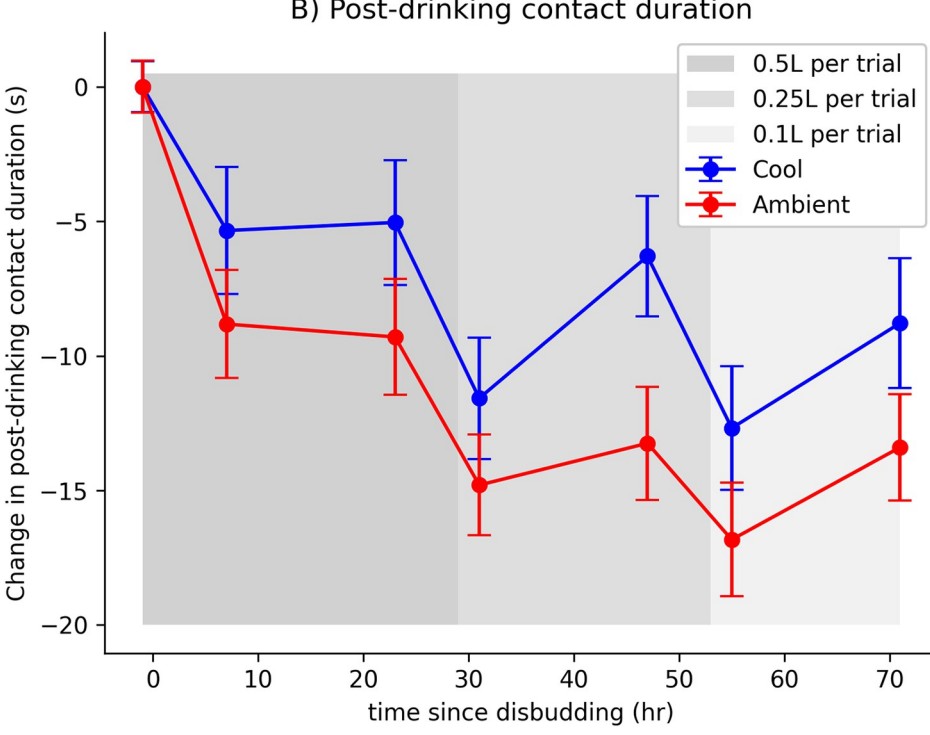

**Fig 3. Changes in response variables post disbudding.** Panel A) shows the change in latencies (from calf-specific baseline session 5) to approach cold or ambient packs as time since disbudding increases. Panel B) shows the change in post-drinking contact duration against time since disbudding. Error bars in each plot represent standard errors. In both plots, data-points at -1h post-disbudding represent session 5 (baseline).

for latency (P = 0.573), drinking duration (P = 0.713), post-drinking contact duration (P = 0.846) or nipple contacts (P = 0.831).

Colour of the pen was found to be significant in only one of the models, with the blue side associated with an increased drinking duration (t = 5.276, P <0.001). To investigate whether the two milk buckets used on each side of the test area emptied at the same rate, a permutation test with 5,000 repeats was conducted between all drinking duration values obtained for the Blue and White pens. On average, the two buckets were not found to empty at different rates (permutation test, P = 0.088), suggesting that the emptying rates of the two buckets does not help explaining the differences in colour.

Although the graphical analyses of variation between each trial revealed no discernible trend for post-drinking contact duration, latency varied across the three trials (Fig 4). No difference in latency between cool and ambient packs for trial 1 can be observed, but the latency to go to the ambient pack during trials 2 and 3 seemed to increase from 23hr post-disbudding (session 7), before returning towards baseline levels after approximately 55h.

## Conditioned place preference

Calves did not spend significantly more time in the pen where they received the cool pack (mean ± SD: 48.96 ± 23.91; one-sample t-test, $t_{13}$ = -0.18, P = 0.864), indicating an absence of conditioned place preference (Fig 5). Initial choice of pen to enter was also non-significant ($\chi^2$ (1, N = 14) = 0.29, P = 0.593).

## Discussion

After disbudding, calves were relatively faster to approach a milk reward associated with application of a cool pack on their head in comparison to the ambient temperature pack.

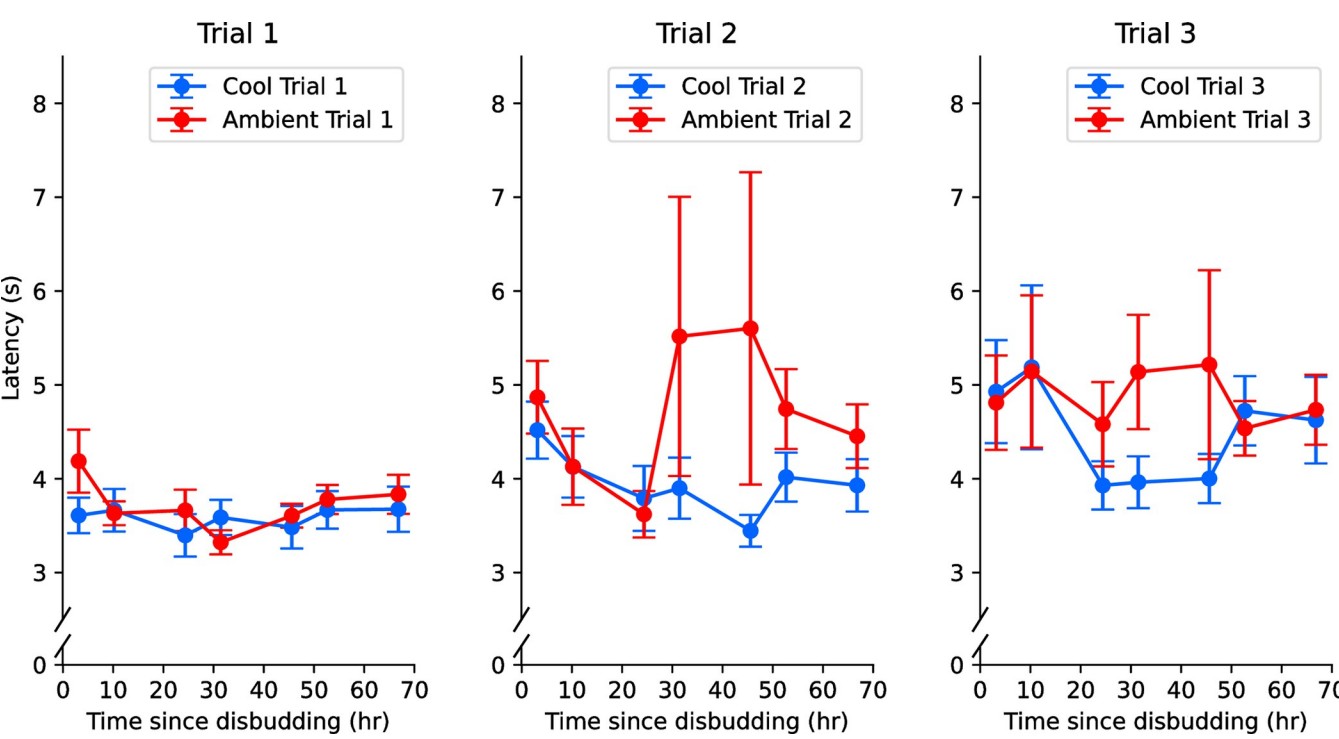

**Fig 4. Latency by session for each trial.** Trial 1 shows no difference between treatments, whilst trial 2 and trial 3 demonstrate a marked increase in average latency from session 7 onwards for the ambient pack only. Session 7 occurs approximately 25h post-disbudding, which corresponds roughly to the duration of effect of the meloxicam anaesthetic [38]. Error bars in each plot represent standard errors.

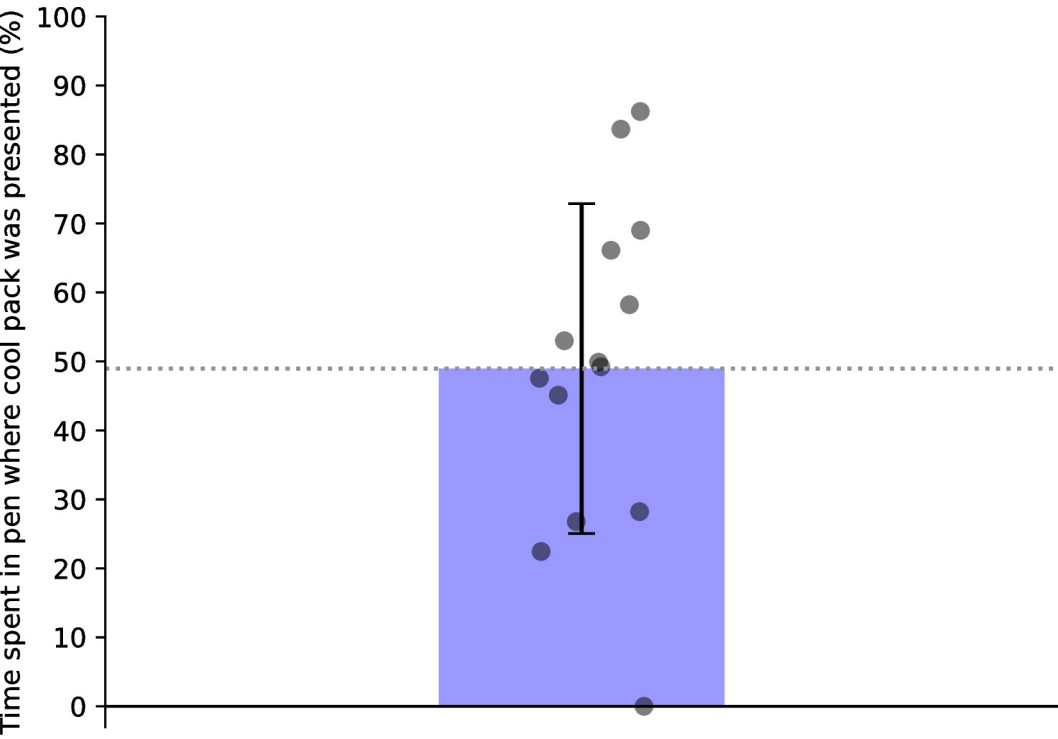

**Fig 5. Plot of conditioned place preference.** Percentage of time spent within the pen associated with the cool pack. A conditioned place preference test was conducted to see whether calves preferentially spent more time in the side associated with the application of the cool pack. Calves were allowed 15 minutes free access to both test pens with the door between them open, following the conclusion of the last approach-aversion test on Saturday morning. Time spent in each half of the test pen was recorded over the 15 minute duration. Error bar represents standard deviation.

Additionally, disbudded calves spent a greater proportion of their time in contact with the cool pack once they were done drinking the reward. In particular, notable declines in latency from the baseline with the cool pack began approximately 24 h post-disbudding, coinciding with the reported duration of action of the analgesic administered during the procedure (meloxicam) [39]. Interestingly, an initial rise in latency to the ambient pack within the first 24 h post-disbudding was not observed for the cool treatment, pointing to a potential benefit of cold therapy also within the first 24 h post-disbudding. These findings support our prediction that application of cold to wound sites post-disbudding provides pain relief.

Whilst the relative post-drinking contact duration was significantly longer for the cool pack compared to the ambient control, all average durations were substantially lower than the pre-disbudding baseline. This sustained drop in contact duration may be a result of multiple factors, including imperfect block [11], post-operative pressure sensitivity [12], avoidance of people associated with prior negative experiences (such as contact with the disbudding wound) [40] or post-procedural anhedonia (deficit in appreciation of reward) and pessimism (lower expectation of positive reward) [8, 26].

Regardless of treatment, calves tended to make fewer nipple contacts, drink slower and maintain pack contact post-drinking for less time as the reward volume was decreased. Whilst trial 1 revealed no observable difference between treatments, trials 2 and 3 illustrated a marked (though temporary) rise in latency for only the ambient pack from approximately 24 h onwards, with the greatest discrepancies between treatments occurring 30–50 h post disbudding. The lack of divergence in trial 1 suggests that first contact with the pack within each session serves as a 'reminder' trial, as calves prior to first pack contact behave identically across

treatments. This suggests that calves might have been unable to assimilate the association between colour ('Blue' vs. 'White') and treatment ('Cool' vs 'Ambient'), instead relying on working memory to anticipate treatment in trial 2 and 3. This could explain the non-significance of expected milk volume detected in the latency model, as well as the lack of effect detected in the CPP test. Alternatively, hunger in trial 1 (but not trials 2 and 3 once they become more satiated) may have been sufficient to hide any influence of treatment, given how low the average latency was. These between-trial observations may explain why the CPP test did not find a significant preference between the two sides of the test pen. No between-trial discrepancy was observed for post-drinking contact duration. If the trial differences in latency did indeed occur because calves needed a trial 1 reminder of the packs' effects, this would explain why contact duration, measured subsequent to calves' first contact with the pack in trial 1, didn't differ across trials.

An additional limitation was running the CPP test following the completion of the approach-aversion experiment, after sessions where the milk reward had decreased. We chose not to conduct CPP prior to reward diminution, so as to not jeopardize the learning for the approach-aversion task. This may also explain why we did not detect differences in the CPP test.

The finding that colour had a significant effect both pre- and post-disbudding is surprising. Whilst the experiment was counterbalanced through pseudorandomised allocation of treatment and colour, small imperceivable differences between the two sides of the test pen may have resulted in the observed significance.

Calves were disbudded at a mean of 58 days of age (approximately 8 weeks). Although most recommendations cite 8 weeks as an upper limit for calf disbudding, this is not based on empirical evidence, instead based on the physiological timeline of underlying periosteum development [41]. Evidence suggests that horn size, rather than age, should be used primarily when making recommendations for time of disbudding, as this accounts for variability in breed, sex and growth [42]. Equally, disbudding too young has shown to have negative consequences on calf pain sensitivity in the weeks after disbudding [3, 20]. Whilst we were limited in our ability to control age at disbudding, all calves enrolled in the study were deemed suitable for disbudding by a qualified veterinarian.

Calves displayed substantial individual differences in behaviour and approach to the task. The influence of inter-individual differences on learning capacities has previously been studied in a wide range of taxa, including invertebrates, fish, birds and mammals [43–46]. Individual personality constitutes a number of important traits such as pessimism and nervousness–which in turn interact with the contextual environment of a performance task to influence cognitive ability, memory and mood–resulting in variable outcomes in learning performance [47–51]. In particular, differences in calf and cattle personality influencing their feeding behaviour have been found to account for nearly 75% of variation in task performance [52, 53]. Variation in individual personality may therefore explain why the observed effect size is somewhat smaller than had been predicted. Additionally, the consistent aversiveness of physical pack contact and human interaction [12, 40] may have prevented the development of pain relief through prolonged contact in some calves. The lack of calf agency over pack application in terms of pressure, duration and temperature might have negatively affected their willingness to engage with cold therapy [8, 9, 54]; the main limitation of this study being human involvement. Further investigation to understand the benefit of post-procedural cold therapy in calves is necessary, possibly through the autonomous application of cool packs or other cooling mechanisms to limit human involvement and biases. While the development of polled (hornless) breeds may serve as a lasting solution to the welfare concerns of disbudding [55], many popular breeds are still disbudded. As such, a better understanding of the nature of post-

procedural pain and the development of low-cost, low-effort solutions of providing pain relief is still of great interest with respect to calf welfare. With the current low levels of NSAID use on farms during disbudding procedures [17], additional research is also required before cold therapy can be practically applied on farms.

## Conclusion

Calves are motivated to seek cold therapy following disbudding, providing evidence of an analgesic effect. This is observed through decreasing approach latencies compared to ambient control packs, as well as a greater persistence to seek contact with the cool pack following consumption of milk rewards. Motivation is most evident in the time period following the loss of systemic pharmacological analgesia and persists for up to 72 hours post-disbudding. However, calves did not develop a conditioned place preference for the pen associated with cold therapy. We suggest cold therapy as a promising method to assess pain associated with disbudding and encourage further research of its use including potential practical applications.

## Supporting information

**S1 File. Dataset.**
(XLSX)

**S2 File. Code used for data transformation and statistical analysis.**
(ZIP)

**S1 Video. Experimental procedure and range of calf responses.**
(ZIP)

**S1 Table. Latency post-disbudding model comparison between inclusion and exclusion of 30s ceiling values.**
(DOCX)

## Acknowledgments

We would like to thank Jillian Hendricks, Juliette Ragot and Natalia Wazny for their help running the experiments at the farm. Further thanks go to the staff at Langford Vets Farm Animal Practice, the John Oldacre Centre, and the farm staff at Wyndhurst farm for their help during the study.

## Author Contributions

**Conceptualization:** Kane P. J. Colston, Thomas Ede, Benjamin Lecorps.

**Data curation:** Kane P. J. Colston, Benjamin Lecorps.

**Formal analysis:** Kane P. J. Colston, Thomas Ede, Michael T. Mendl, Benjamin Lecorps.

**Funding acquisition:** Thomas Ede, Michael T. Mendl, Benjamin Lecorps.

**Investigation:** Kane P. J. Colston, Benjamin Lecorps.

**Methodology:** Kane P. J. Colston, Benjamin Lecorps.

**Project administration:** Benjamin Lecorps.

**Resources:** Benjamin Lecorps.

**Supervision:** Thomas Ede, Michael T. Mendl, Benjamin Lecorps.

**Validation:** Thomas Ede, Michael T. Mendl, Benjamin Lecorps.

**Visualization:** Kane P. J. Colston, Thomas Ede, Michael T. Mendl, Benjamin Lecorps.

**Writing – original draft:** Kane P. J. Colston.

**Writing – review & editing:** Kane P. J. Colston, Thomas Ede, Michael T. Mendl, Benjamin Lecorps.

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
