## [Editor Report · Decision Letter 0]

21 Nov 2023

PONE-D-23-37809Cold therapy and pain relief after hot-iron disbudding in dairy calvesPLOS ONE

Dear Dr. Lecorps,

Thank you for submitting your manuscript to PLOS ONE. After careful consideration, we feel that it has merit but does not fully meet PLOS ONE’s publication criteria as it currently stands. Therefore, we invite you to submit a revised version of the manuscript that addresses the points raised during the review process. *See Editor Comments below*

We look forward to receiving your revised manuscript.

Kind regards,

Angel Abuelo, DVM, MRes, MSc, PhD, DABVP (Dairy), DECBHM

Academic Editor

PLOS ONE

Journal Requirements:

2. To comply with PLOS ONE submissions requirements, in your Methods section, please provide additional information regarding the experiments involving animals and ensure you have included details on methods of anesthesia and/or analgesia

   "This research was funded by the Animal Welfare Research Network’s Kick-start funding award and the University of Bristol through the start-up research funds attributed to Benjamin Lecorps. "

Additional Editor Comments:

Before this manuscript can be sent out for peer-review, please revise the manuscript addressing the following:

- Using correct epidemiological terms, define the type of study design utilized. Then, follow the relevant reporting guidelines. Currently, many essential information that must be reported is missing. When resubmitting, please include the checklist of the appropriate reporting guidelines outlining where in the manuscript you are reporting each item.

- Provide the specific details of the sample size calculation. Simply stating "we used previous peer-reviewed publications with similar methodologies as a basis for our sample size calculation" is inadequate.

---

## [Author Response · Author response to Decision Letter 0]

27 Nov 2023

Please note that a response to the editor's comments has been uploaded.

Authors’ response

We have listed comments from the editor (in red) followed by the measures we took to address them (in black, preceded by >>>), along with manuscript quotations where appropriate. Page/line numbers given in responses refer to the ‘Tracked_changes’ version of the revised manuscript (file name: “Revised Manuscript with Tracked Changes”).

Responses to Editor:

>>> Manuscript author affiliations, figures and headings have been edited in accordance with the guidance.

2. To comply with PLOS ONE submissions requirements, in your Methods section, please provide additional information regarding the experiments involving animals and ensure you have included details on methods of anaesthesia and/or analgesia

>>> We are unsure of any additional information that could be provided regarding experimental designs which is explained in length in the Methods section. Methods involved with respect to anaesthesia and analgesia are already detailed: 

Line 147-148: “ Calves were administered a subcutaneous NSAID analgesic (meloxicam, 0.5 mgkg-1) and a procaine cornual nerve block (160 mg per horn bud), five minutes prior to disbudding.” 

We are happy to address further specific concerns the editor may have.

>>> The funding statement has been removed from the main manuscript.

 "This research was funded by the Animal Welfare Research Network’s Kick-start funding award and the University of Bristol through the start-up research funds attributed to Benjamin Lecorps. "

>>> The role of funder statement has been included within the funding statement and in the cover letter. 

>>> The minimal underlying dataset was uploaded as Supporting Information file (S1) (pg 18) when submitted initially and as indicated during the submission process. We now make this clear in the cover letter as well.

>>> Supporting Information has been updated accordingly (pg18).

Additional Editor Comments:

Before this manuscript can be sent out for peer-review, please revise the manuscript addressing the following:

- Using correct epidemiological terms, define the type of study design utilized. Then, follow the relevant reporting guidelines. Currently, many essential information that must be reported is missing. When resubmitting, please include the checklist of the appropriate reporting guidelines outlining where in the manuscript you are reporting each item.

>>> As this is not an epidemiological study (but rather an experimental one), we are unsure on how best to describe the study design. Can the editor be more specific about the ‘many essential information’ required that would be missing? This is not the first time we submit to PLOS One and this manuscript is not less complete than any of our previous ones. Please note that we have included in our submission a copy of a completed ARRIVE Essential 10 guideline, with line number corresponding to the updated manuscript. 

- Provide the specific details of the sample size calculation. Simply stating "we used previous peer-reviewed publications with similar methodologies as a basis for our sample size calculation" is inadequate.

>>> We have addressed concerns regarding the calculation/justification of our sample size. Because no previous studies combined approach avoidance and conditioned place avoidance tests before, nor assessed the effect of cold therapy to study pain in this animal species or in any other, we could not directly use previous effect sizes for a priori power calculations. This means we had to rely on estimated effect sizes.

Line 82: “Using the pwr package in R [33], a minimum sample size of 17 individuals was calculated for a statistical power of 0.8, a small effect size (f2=0.02) and a significance level of 0.05 using linear models for our two main variables (latency and post-drinking contact duration in the approach-avoidance test). A similar power calculation for t-test found that 15 individuals would be necessary to detect strong effect size, consistent with previous studies...”

>>> all figures have been run through PACE and are in a .tif format.

---

## [Decision Letter · Decision Letter 1]

1 Mar 2024

PONE-D-23-37809R1Cold therapy and pain relief after hot-iron disbudding in dairy calvesPLOS ONE

Dear Dr. Lecorps,

Thank you for submitting your manuscript to PLOS ONE. After careful consideration, we feel that it has merit but does not fully meet PLOS ONE’s publication criteria as it currently stands. Therefore, we invite you to submit a revised version of the manuscript that addresses the points raised during the review process. As per previous correspondence, there were some discrepancies between the comments of the 2 initial reviewers, and a 3rd reviewer was enlisted. Thank you for your patience in receiving an editorial decision. Collectively, there are some important concerns regarding the study design and interpretation of results. I invite you to address all reviewers' concerns by revising your manuscript accordingly.

We look forward to receiving your revised manuscript.

Kind regards,

Angel Abuelo, DVM, MRes, MSc, PhD, DABVP (Dairy), DECBHM

Academic Editor

PLOS ONE

**Additional Editor Comments:**

*In the manuscript, please specify the type of study using adequate terminology (e.g., case-control, cross-sectional, cohort, randomized control trial, etc.)*

Reviewers' comments:

Reviewer's Responses to Questions

**Comments to the Author**

1. If the authors have adequately addressed your comments raised in a previous round of review and you feel that this manuscript is now acceptable for publication, you may indicate that here to bypass the “Comments to the Author” section, enter your conflict of interest statement in the “Confidential to Editor” section, and submit your "Accept" recommendation.

Reviewer #1: (No Response)

Reviewer #2: (No Response)

Reviewer #3: (No Response)

2. Is the manuscript technically sound, and do the data support the conclusions?

Reviewer #1: Partly

Reviewer #2: Yes

Reviewer #3: No

3. Has the statistical analysis been performed appropriately and rigorously? 

Reviewer #1: I Don't Know

Reviewer #2: Yes

Reviewer #3: Yes

4. Have the authors made all data underlying the findings in their manuscript fully available?

Reviewer #1: Yes

Reviewer #2: Yes

Reviewer #3: Yes

5. Is the manuscript presented in an intelligible fashion and written in standard English?

Reviewer #1: Yes

Reviewer #2: Yes

Reviewer #3: Yes

6. Review Comments to the Author

Reviewer #1: This paper evaluates the use of "cold therapy" as a mechanism to reduce pain associated with dehorning. There are several challenges associated with the description of the study that makes it difficult to interpret the results. Furthermore, disbudding was completed very late and many different breeds were used. The practical relevance of producers using cold therapy is also a concern. Specific comments can be found below:

Line 20: Please clarify the type of analgesia provided

Line 21: It is unclear from the abstract how frequently the "cool packs" were changed or how they were placed on the calf's head and for how long. These seem like critical details to assess the practicality of this approach working.

Line 23 to 24: The calves were housed in completely separate pens when the treatments were being applied? This seems like it could introduce substantial bias as it would be impossible to control for the impact of pen or calves within the pen.

Line 39 to 41: I think it is important to note that the use of pain mitigation is low in some countries. For example, in Canada, the overwhelming majority of producers use pain control (it is a requirement), whereas in the US, most use pain control as well. This brings up another point that is worth discussing in the discussion of this paper. If few producers are using pain control, how will we convince them to use cold therapy which would seemingly be harder to manage?

Line 56 to 58: How was it determined to be beneficial? Through preference testing?

Line 85 to 86: Above, it mentions few have done this type of work making it difficult to calculate; however, here it suggests previous studies are done? Basing sample size on simply what other studies have enrolled is likely inappropriate and should be deleted. It is different to use effect sizes found in the studies to come up with a calculation.

Line 87 to 88: I assume that breeds would not be balanced between groups making it difficult to control for the likely effect that breed could have on the testing done in this study. How was that controlled for? If not controlled for, this should be noted as a limitation.

Line 88: When were calves normally weaned at this facility? Why was disbudding completed so late? Earlier disbudding would reduce inflammation and the size of the lesion. Likely important to preface the findings with the dehorning time in mind.

Line 92: Additional information is needed on the feeding program provided ahead of time.

Line 93: 190 g of milk replacer per L is very concentrated. Is this correct?

Line 94 to 95: Were groups made up to contain both treatment groups?

Line 110: How was this temperature maintained?

Line 118: Was the cold pack only applied for 20 min (or seems like 40 min per day for 6 days) and no longer? How effective is short application of cold packs in controlling inflammation? Is there any data or studies that could be cited here to provide insight into why this threshold was chosen?

Line 125: How was this milk prepared (temperature and g/L)? Was it the same milk replacer? Was time relative to last milk replacer meal controlled for? i.e., if a calf was fed milk at 8 am, was it tested at 8:30? Similarly, if a calf was fed at 8 would it be tested at 13:30? This would have a major impact on milk consumption.

Line 147 to 148: Pleas add in the brand of meloxicam and procaine used.

Line 155: This is confusing. 3.75 L were consumed per session? I assumed it went down as highlighted in Line 151 to 153. Please clarify

Line 164 to 165: Why were they excluded?

Line 171: Why are these excluded?

Line 191: How were these models evaluated to ensure model fit?

Line 233: Please provide information about the comparability of the two treatment groups including age at disbudding and breed to ensure comparable between groups.

Line 241: Pack temperature or treatment group?

Line 259: Was coloured controlled for in the model as well?

Line 300 to 301: What does "near-significant" mean? This is also not in the results as far as I can tell.

Line 314 to 317: I find this very confusing. Could a different word be used instead of trial? When were the calves disbudded and how did time relative to disbudding come into play?

Reviewer #2: This study uses approach-avoidance as well as Conditioned place preference to evaluate the effect of a cool pack on post disbudded calves. 15 calves were enrolled prior to the disbudding procedure and habituated and conditioned to reception of a cool or ambient pack while drinking milk. Calves were disbudded with local and systemic anesthesia and analgesia. The authors found that calves had a reduced latency to approach when a cool pack was offered vs. an ambient temp pack, but that calves did not display a preference to the pen where they received the cool pack.

This is a very interesting study that pairs two tests to evaluate the emotional component of pain/pain relief. The study is well written.

Comments:

L41. Is disbudding a procedure that has to be performed by a veterinarian in the UK? Is that why “workload” is cited as a reason to not provide pain mitigation? Please clarify.

L58. In that study, calves at 20d all still had necrotic buds, so is it still pain? What about sensitivity? Itchiness due to healing epithelial tissue?

L88. These calves are quite old to disbud (with most (Holstein) recommendations begin to disbud younger than 4 weeks of age). Why was the procedure done so late?

L128: Videos didn’t work for me, which was disappointing!

L134: So each calf received each pack/color combination (which stayed the same so if calf 1 had blue/cold they always got blue/cold)? Thus if that calf wanted the cold pack after disbudding, they would have to go to blue? Why not apply the cold pack to the calf post disbudding for some period of time paired to color? Couldn’t it be that cold on a “normal” calf head isn’t the same as cold on a disbudded head (soothing the inflammation, for example) and that you would have had a stronger response? Please clarify in the methods.

L343. Yes, would have been very interesting to give the calf the choice (milk meal paired with color/pack) but that would be much more difficult to habituate/train?

L346. Consider adding some discussion of calf age and how age (and bud size) might impact the calf response to this type of treatment. Would suspect that younger calves (shorter burn time) might not respond the same way? Any more information available on technique used in this study? Time of procedure (how long to burn?)?

Figure 4. Confused by the Trial 1/Trial 2/Trial 3…..what are these? There is no reference to these three trials anywhere else in the manuscript (that I can find). Please clarify.

Reviewer #3: I commend the authors for their creative approach to attempt to offer calves autonomy and additional pain relief given the long-lasting pain from disbudding. I appreciate the acknowledgement of the limitations for this study, such as the motivation for milk influencing the latency to respond in Trial 1. However, there were several conceptual areas that I found concerning or where my interpretation is that additional information would be useful.

1.) The use of an approach/aversion paradigm and conditioned place preference/aversion are not consistently described throughout the manuscript. I would benefit from some clarification and consistency.

o Ln 67 lays out the test as an approach-aversion test, while the predictions do not specify which aspect of the treatments would be aversive (ln 70-72).

o In Ede et al 2018 (10.1038/s41598-018-27669-7), the treatment was something the authors predicted that the calves would avoid (injection pain), thus naming it an approach/avoid paradigm in their work. As far as I can tell, in the current study, the prediction was that the calves would like the cold (ln 70-72), thus it becomes an approach/approach paradigm, which is more difficult to untangle in terms of what it means to the calf than what Ede et al 2018 did. Consider discussing this.

o It would be reasonable to predict that calves would find either of the packs aversive given the increased sensitivity following disbudding. This is touched on this briefly in ln 310. If you choose to continue describing the test as an approach-aversion test, I would benefit from some context for the aversive component in the introduction. In addition to references 12, 20, and 21, you may consider discussing other literature that provides evidence of sensitivity to touch following disbudding, to support the aversive component of your test (e.g.: 10.3168/jds.2021-21552; 10.3168/jds.2023-23238; 10.3168/jds.2009-2813).

o Ln 177-181 begins by describing the use of a conditioned place preference test, but ends by stating calves were tested for aversion. Ln 281-284 and Figure 5 present the preference of calves for the cold pack, but do not report any conditioned place aversion. Please clarify if conditioned place aversion was also tested or observed.

2.) The conclusions drawn are overstated and do not seem supported by the approach or data presented

o Ln 350-351: I do not see evidence presented in the current paper of the analgesic effects of cold therapy for calves. Rather, calves seek a milk reward faster with a cold pack is applied to their head compared to an ambient-temperature pack.

o Ln 26-27, 306: I do not see evidence that “suggests the cold therapy provides some form of pain relief post-disbudding” Rather, calves seek a milk reward faster with a cold pack is applied to their head compared to an ambient-temperature pack. Pain relieving characteristics were not evaluated.

o Ln 346: Further work is needed to understand possible benefits of cold therapy, rather than “confirm”.

o As the cold therapy was tied to a milk reward, it is difficult to disentangle the effects of the cold therapy from their motivation for milk. With the current methods, calves will likely approach the packs unless they are not hungry, which seems unlikely at this plane of nutrition and age of the calves. I would benefit from further discussion of the implications of this context when assessing the results.

3.) Use of the milk reward in conjunction with the compress.

o In Ln 92, it seems calves being provided their milk meals only in the arena? Was there any milk meals provided outside of the training or test sessions?

o Ln 154-155 was the remaining milk replacer fed in different location? Was this changed in any way as the reward size changed?

o Ln 187-188 given the ages, were any calves being weaned during the trial?

o The drinking speed seems rather fast (0.5L/60 sec; ln 151, 171), I would benefit from some contextualization of these speeds, how they compare to other literature, and biologically normal drinking behavior.

o What are the possible implications of initiating the suckling response to then end a training session and restart? For more context about how milk stimulates sucking, see, for example, de Passillé 2001, Sucking motivation and related problems in calves, 10.1016/s0168-1591(01)00108-3

4.) What do we know about cold therapy?

o I would benefit from additional information on the use of cold therapy.

o How long is cold therapy needed to receive the benefits of pain relief? Is it possible to provide context of timing in humans and other species?

o I appreciate that citations 29 and 30 are reviews of this topic, is there specific evidence that can be pulled to help provide additional context for this pain relief strategy?

o Is there evidence that the cold therapy actually cooled the wounds? Surface temperature of the calf’s head at the start vs after pack application?

5.) Why were the calves/trials where milk drinking persisted for longer than one minute excluded?

o What was the number of calves effected by each exclusion of data? How did those relate to treatment (e.g., Ln 165, 171)?

o The exclusion of these trials or limiting the compress contact to ~2 minutes seems to assume that the benefits from the cold are less than 60 seconds. Is there evidence to support this assumption?

6.) Additional points that I would benefit from clarification on:

o I am struggling with some of the mismatch in language used throughout the paper, particularly when discussing different metrics to assess calf welfare. For example, in ln 51, it is said that behavioral and physiological evidence provide weak evidence for animal’s emotional state, but later in the paper, there seems to be strong conclusions drawn using behavioral metrics. e.g., ln 299-300 conclusions about these responses and how they link to emotional state are made.

o There seems to be a discrepancy in the amount of discussion among results, where some considerations that are only tangential given much more space than those that are concrete and connected to results shown. For example, there a limited amount of discussion on the effects of milk allowance (ln 300-303; this is a concrete, important consideration to understand the findings), but other components that were not reported in this study, such as personality, were discussed in length (ln 333-341).

o Ln 109- 111: How was temperature maintained for the cold packs? Including this information would be helpful for others aiming to replicate your work

o Ln 146-147: Consider including details of the disbudding process, for example, the iron used and if the bud was left in or scooped out

o Ln 162-163: Could you provide an operational definition for “gazing” as part of your definition for latency to drink? How is this assessed for an animal with approximately 270 degrees in their visual field?

o Ln 199-201: Why were the data transformed through the addition of a constant to eliminate negative values? Given the interest in change from baseline, wouldn’t the negative values have meaning?

o Ln 56-58: What is the benefit of phrasing the study results in this way as opposed to contextualizing CPP, which is a component of your research?

o Ln 156-160: Why choose to conduct the CPP at the end of testing sessions when milk has been decreased rather than after session 7? Could this affect the association the calves would have made in the sessions running up to the CPP?

o Ln 87-88: Did you investigate any effects of breed?

o Ln 122-123: Was the side of the pen the color was on also altered? Or was blue always left as depicted in Figure 1?

7. PLOS authors have the option to publish the peer review history of their article (what does this mean?). If published, this will include your full peer review and any attached files.

Reviewer #1: No

Reviewer #2: No

Reviewer #3: No

---

## [Author Response · Author response to Decision Letter 1]

31 Mar 2024

Line numbers in the following responses to comments refer to the revised manuscript without tracked changes. 

Editor Comments:

In the manuscript, please specify the type of study using adequate terminology (e.g., case-control, cross-sectional, cohort, randomized control trial, etc.)

>>> We have included this in our introduction (line 71). it now reads:

“For this we conducted an experimental control trial, using a methodology combining an approach-aversion test and a conditioned place preference (CPP) procedure.”

Reviewers' comments:

Reviewer #1: 

This paper evaluates the use of "cold therapy" as a mechanism to reduce pain associated with dehorning. There are several challenges associated with the description of the study that makes it difficult to interpret the results. Furthermore, disbudding was completed very late and many different breeds were used. The practical relevance of producers using cold therapy is also a concern. 

>>> Thanks for your comments. We designed this experiment as a proof of concept. Our aims were to assess the possibility of providing pain relief using cold therapy so that we can then use calves’ preference or willingness to work for this to assess for how long calves are in pain following disbudding. This knowledge will then inform pain mitigation strategies. Of course, an application of cold therapy to provide pain relief post-disbudding may derive from this work but this was not the primary purpose of this paper.

Specific comments can be found below:

Line 20: Please clarify the type of analgesia provided

>>> abstract edited to include meloxicam – line 20 now reads “Calves were then disbudded under local anaesthesia (procaine) and analgesia (meloxicam) ”.

Line 21: It is unclear from the abstract how frequently the "cool packs" were changed or how they were placed on the calf's head and for how long. These seem like critical details to assess the practicality of this approach working.

>>> the abstract has been reworded to address the concerns raised. We are however restricted with abstract word limits, and the details regarding the nature of pack contact are elaborated upon in the methodology. The abstract now reads (line 18-20): “Calves were habituated to the manual placement of a cool or ambient pack on their forehead for a short duration simultaneous to milk reward consumption, prior to disbudding.”

Line 23 to 24: The calves were housed in completely separate pens when the treatments were being applied? This seems like it could introduce substantial bias as it would be impossible to control for the impact of pen or calves within the pen.

>>> Sorry for the confusion. Calves were housed for the entire duration of the study in a familiar hutch (described in the ‘animals and housing’ section). During experimental sessions, calves were temporarily moved to the experimental arena (figure 1) – this experimental arena was the same for every calf tested. Each calf received the two different treatments (cool pack application and ambient temperature control pack application) in different sides of the colour-coded experimental pen (figure 1), but the pseudorandomised counter-balancing of treatment and colour allowed us to control for the impact of pen side (i.e. half the calves receive the cool pack on the blue side and ambient on the white, the other half of the calves this is reversed). All experiments were conducted with calves one at a time, so no influence of other calves in the experimental pen needs to be considered. 

To try and make this clearer in the abstract, lines 21-23 have been changed to: “Individual calves were consistently exposed to either cool or ambient packs in different halves of a two-sided experimental pen, allowing for the testing of approach-aversion and conditioned place preference”

Line 39 to 41: I think it is important to note that the use of pain mitigation is low in some countries. For example, in Canada, the overwhelming majority of producers use pain control (it is a requirement), whereas in the US, most use pain control as well. This brings up another point that is worth discussing in the discussion of this paper. If few producers are using pain control, how will we convince them to use cold therapy which would seemingly be harder to manage?

>>> This is an interesting comment regarding the promotion of changes in pain-management practices. As explained in response to a previous comment, the aim of this study was not to explore the applicability of cold therapy to mitigate pain associated with cold therapy on a farm. 

Line 56 to 58: How was it determined to be beneficial? Through preference testing?

>>> This paragraph has now been reworded for clarity. It now reads (line 55-61): 

“Conditioned place preference (CPP) tests – a commonly used paradigm to assess the rewarding effects of a specific stimulus or experience as a result of prior conditioning [28] – have also indicated that post-procedural pain and its associated emotional impact are remembered by calves [7]. Using the CPP paradigm, calves have also shown preference for the provision of analgesia with disbudding [29], as well as prolonged pain control (i.e., in the form of a local anaesthetic injection) up to 20 days after disbudding when compared to sham disbudded controls [14], suggesting that they were still in pain.”

Line 85 to 86: Above, it mentions few have done this type of work making it difficult to calculate; however, here it suggests previous studies are done? Basing sample size on simply what other studies have enrolled is likely inappropriate and should be deleted. It is different to use effect sizes found in the studies to come up with a calculation.

>>> Perhaps the reference to ‘previous studies’ has been misleading here – this specific methodology is novel and as such power calculations could not be drawn directly from previous effect sizes. Instead, we used the sample size and effect size of previous experiments conducted on calf responses to pain that use the approach-avoidance paradigm, as this is the closest we are able to get to direct experimental comparisons. Whilst not ideal, this was the optimal approach to determining a sample size when no similar work has been conducted. 

In light of this feedback, we have amended this sentence (line 96-97): “ …consistent with previous studies that have also used an approach-avoidance paradigm to investigate calf response to painful stimuli [29], [36]”

Line 87 to 88: I assume that breeds would not be balanced between groups making it difficult to control for the likely effect that breed could have on the testing done in this study. How was that controlled for? If not controlled for, this should be noted as a limitation.

>>> This study did not aim to assess whether calves of different breed would react to disbudding in different ways. Although we agree no conclusions can be drawn on breed effect, we disagree that having multiple breeds is a weakness, especially when considering that we used within-calf comparisons. In contrast, this is likely to render our results more applicable and translatable to what happens on farms (given that crossbreeds are typically disbudded). We also call the reviewer’s attention to the risks associated with over-standardization and the benefits to introduce controlled variation in experimental designs (see the work of Hanno Wurbel’s team, for example: https://doi.org/10.1038/s41583-020-0370-7). The methodology of our experiment as a within-subject design means breed was controlled for in this manner.

Line 88: When were calves normally weaned at this facility? Why was disbudding completed so late? Earlier disbudding would reduce inflammation and the size of the lesion. Likely important to preface the findings with the dehorning time in mind.

>>> Calves are typically weaned at 12 weeks of age at this farm. We can only speculate about farm staff’s decision and rationale, but we think it was related to the perception that young calves are more vulnerable and therefore should not be disbudded. As researchers, we had limited control over when disbudding was carried out for this experiment. If that can be of any reassurance, calves’ horns were not developed, and the procedure could still be considered disbudding. Furthermore, Marquette et al. (2021) (https://doi.org/10.1186/s13620-021-00196-0) suggests horn size is more relevant than age regarding recommendations for time of disbudding, and Mirra et al. (2018) (https://doi.org/10.1016/j.physbeh.2017.11.031) shows that age at disbudding does not affect pain sensitivity. Considering there is no clear-cut distinction between disbudding and dehorning and that age is not the only factor at play, we would prefer to keep the language consistent here and avoid bringing some confusion by talking about disbudding vs dehorning. 

We have added the following paragraph to the discussion section to clarify the concerns raised (lines 367-375):

“Calves were disbudded at a mean age of 58 days (approximately 8 weeks). Although most recommendations cite 8 weeks as an upper limit for calf disbudding, this is not based on empirical evidence, instead based on the physiological timeline of underlying periosteum development [42]. Evidence suggests that horn size, rather than age, should be used primarily when making recommendations for time of disbudding, as this accounts for variability in breed and sex at a given age [43]. Equally, disbudding too young has shown to have negative consequences on calf pain sensitivity in the weeks after disbudding [3], [21]. Whilst we were limited in our ability to control age at disbudding, all calves enrolled in the study were deemed suitable for disbudding by a qualified veterinarian.”

Line 92: Additional information is needed on the feeding program provided ahead of time.

>>> The feeding regime prior to the experiment with the calves was identical to what was received during the experiment (ad libitum access to water, grain and straw, plus 7.5L milk per day). For clarification, the section under ‘Animals and housing’ now reads (line 101-105):

“Calves were given ad libitum access to water, grain, and straw, as well as a total of 7.5L of milk replacer per day during the experimental sessions (Sprint Plus 50, Bridgmans Farm Direct; 150gL-1), split equally between experimental sessions in the morning (0800 h) and afternoon (1600 h). This is identical to the feeding program delivered to calves prior to experimentation.”

Line 93: 190 g of milk replacer per L is very concentrated. Is this correct?

>>> This was a typo. The concentration was 150gL-1, and this has been corrected in the manuscript. 

Line 94 to 95: Were groups made up to contain both treatment groups?

>>> Every calf experienced both treatments, alternating between cool and ambient pack application on different sides of the pen. This is explained in the ‘experimental procedure’ section, and has been elaborated on in a revised figure 2 (below). 

Line 110: How was this temperature maintained?

>>> Cold packs were kept in a fridge, maintained at 4ᵒC until use. Each pack was swapped for another identical pack at 4ᵒC after each experimental session with a calf. Using an infrared thermometer, we confirmed that over the duration of a session (a maximum of 20 minutes), cool pack temperatures only increased by a maximum of 1ᵒC. 

To clarify, we have added a statement to this effect (line 127-129): “Cool pack temperatures were monitored using an infrared thermometer (SOVARCATE HS960D), confirming that pack temperatures did not change by more than 1ᵒC during a given session.”

Line 118: Was the cold pack only applied for 20 min (or seems like 40 min per day for 6 days) and no longer? How effective is short application of cold packs in controlling inflammation? Is there any data or studies that could be cited here to provide insight into why this threshold was chosen?

>>> This comment is a misunderstanding of our methodology, for which we apologise. Packs were not consistently applied for 20 minute durations, but rather only when calves were drinking milk in a given trial. In a session that typically lasts for a maximum of 20 minutes, 6 trials take place, but the exact duration that time the pack is applied is variable, as it depends on how long the calf chose to suckle in each trial. Furthermore, our intention in this study was not to use cold therapy for its anti-inflammatory properties, but rather for its instantaneous pain relief effects (compare this to the immediate pain relief in humans experienced from cold running water or the application of a cool pack on a burn). 

To help avoid this misunderstanding, we have changed a sentence in the introduction, which now reads (lines 63-68): ““Whilst less regularly used in animals for analgesic purposes, cold therapy is suggested as “…an effective analgesic tool for acute pain management” [32], and appears suitable to cautery disbudding given the nature of the injury associated with the procedure. With running water being likely impractical for the disbudded wound [33], cold spray as a pre-disbudding pain mitigation strategy has been trialled previously in calves indicating an analgesic effect [34].”

We have also made a small edit to the introduction to emphasise that we are investigating instantaneous pain relief, rather than prolonged anti-inflammatory relief, so the final paragraph now starts (69-71): “The aim of our study was to investigate whether cold therapy, through the manual application of a cool pack to the disbudding wound, would be able to provide instantaneous pain relief on the days following the procedure.”

Line 125: How was this milk prepared (temperature and g/L)? Was it the same milk replacer? Was time relative to last milk replacer meal controlled for? i.e., if a calf was fed milk at 8 am, was it tested at 8:30? Similarly, if a calf was fed at 8 would it be tested at 13:30? This would have a major impact on milk consumption.

>>> We added some information (line 105-108):

“Milk replacer was mixed with warm water (45ᵒC) as stated by the provider. Experiments were conducted at the meal times (8am and 4pm), so when calves did not receive their full milk allowance (3.75L per meal) during an experimental session, they were fed the remaining amount just after the session ended.”

Line 147 to 148: Pleas add in the brand of meloxicam and procaine used.

>>> The brand names have been added as requested. the sentence now reads (lines 171-172):

“Calves were administered a subcutaneous meloxicam analgesic (Metacam, 0.5 mgkg-1) and a procaine cornual nerve block (Procamidor Duo,160 mg per horn bud), five minutes prior to disbudding.”

Line 155: This is confusing. 3.75 L were consumed per session? I assumed it went down as highlighted in Line 151 to 153. Please clarify

>>> Sorry for the confusion. See previous comments for clarification.

Line 164 to 165: Why were they excluded?

>>> Trials were ended immediately if latencies exceeded 30s. Although there is always a cost in applying cut-off thresholds, 30s is often used in studies where calves have to make a choice (e.g. Lecorps et al., 2018). We subsequently felt exclusion of ≥30s latencies was the most appropriate approach. Whilst this reduced our data size slightly and prevented us from capturing extremely aversive calf responses, including ≥30s latencies may have had a disproportionate effect in our models. We therefore chose to exclude these latencies from the model. To this effect, the section now reads (line 188-192):

“If the latency for a given trial exceeded 30s, the trial was ended and the calf returned to the lobby area, to commence the next trial, consistent with previous studies [27]. Latencies exceeding 30s were excluded from subsequent statistical analysis (21/588 trials excluded). Whilst cut-off thresholds prevent the capture of extremely aversive calf responses, the exclusion of these latencies ensured our models were not disproportionately influenced by extreme values. 

Line 171: Why are these excluded?

We excluded calves that took greater than 60s to drink for a similar reason – given the rate of consumptions noted previously, we considered 60s to consume the reward to be a sufficiently long time to allow for consumption. A cut-off time was necessary to m

---

## [Decision Letter · Decision Letter 2]

6 May 2024

PONE-D-23-37809R2Cold therapy and pain relief after hot-iron disbudding in dairy calvesPLOS ONE

Dear Dr. Lecorps,

Thank you for submitting your manuscript to PLOS ONE. After careful consideration, we feel that it has merit but does not fully meet PLOS ONE’s publication criteria as it currently stands. Therefore, we invite you to submit a revised version of the manuscript that addresses the points raised during the review process. **We appreciated the thoughtful revisions. Nevertheless, as highlighted by both reviewers, there are still issues pending that need to be addressed before the manuscript could be considered further. Please address these carefully.**

We look forward to receiving your revised manuscript.

Kind regards,

Angel Abuelo, DVM, MRes, MSc, PhD, DABVP (Dairy), DECBHM

Academic Editor

PLOS ONE

Reviewers' comments:

Reviewer's Responses to Questions

**Comments to the Author**

1. If the authors have adequately addressed your comments raised in a previous round of review and you feel that this manuscript is now acceptable for publication, you may indicate that here to bypass the “Comments to the Author” section, enter your conflict of interest statement in the “Confidential to Editor” section, and submit your "Accept" recommendation.

Reviewer #1: All comments have been addressed

Reviewer #3: (No Response)

2. Is the manuscript technically sound, and do the data support the conclusions?

Reviewer #1: Yes

Reviewer #3: No

3. Has the statistical analysis been performed appropriately and rigorously? 

Reviewer #1: Yes

Reviewer #3: No

4. Have the authors made all data underlying the findings in their manuscript fully available?

Reviewer #1: Yes

Reviewer #3: Yes

5. Is the manuscript presented in an intelligible fashion and written in standard English?

Reviewer #1: Yes

Reviewer #3: Yes

6. Review Comments to the Author

**Reviewer #1: **Thanks for your responses to the comments. I understand in science, it is sometimes necessary to "push" the boundaries; however, the applicability of this is limited. This should be discussed especially when producers aren't even doing "easy" methods of pain control (i..e local + NSAID)

Line 19: Please add in the duration of the contact time

Line 95 to 97: I find this revised sentence confusing. I would delete that other studies have used 15 calves previously as it doesn't add meaning

**Reviewer #3: **Thanks to the authors for including the new figure, refining the use and predictions associated with the approach-aversion methodology and clarifying the conclusions about the new information generated about cold therapy. I have one outstanding concern in the R2 version.

I find the approach to excluding data (latency exceeded 30s, drinking exceeded 60s) concerning, as written. I’ve pasted the responses to Reviewers 1 and 3 about this topic below. From what I understand thus far, I am concerned about the approach taken, to exclude these data, has on the conclusions drawn.

A priori or post-hoc decision. Firstly, for the trials ended if latency was greater than 30s and drinking events were capped at 60s, I think it is important to know if this decision was made a priori or post hoc. As the authors point out, these kinds of cutoffs are common when running trial-based experiments. However, if these data were actually collected, then excluding them and imposing a 30s or 60s end was a post-hoc choice, this is worrisome to me. From my perspective, trimming a dataset to remove extreme values post hoc is not best practice.

Alternatives to exclusion. I appreciate that extreme values can affect statistical models, but exclusion should be a last resort. It is entirely possible that these responses represent an important part of the biological response of the calf. As the authors note, these represent “extremely aversive calf responses” and this was part of the question being asked. As such, exclusion is not “taking a conservative approach” as the authors describe it, at least from my perspective it is not conservative. To me, there are several options that would be preferable to the current approach of complete exclusion from a given model. Even when thresholds are set a priori, I prefer to enter a ceiling value for animals that do not meet or exceed a predetermined threshold, for example 31s for latency and 61s for drinking events, in this case (1 s beyond the cutoff). This approach allows the ceiling effect to be included while still meeting some of the assumptions of the statistical tests (e.g. normality). Another option, especially if data were collected beyond the thresholds (that is the decision about thresholds were post hoc) and actual ceilings were reached (e.g. say 80s drinking event), would be to keep all values in the dataset and explore data transformation in parametric models or use of non-parametric tests to address the model fit concerns the authors describe. Lastly, another approach I admire is to present the model outcomes with and without outliers (or extreme values in this case) included. This allows the reader to see their effect for themselves and make their own decision about how much they trust the findings.

Treatment differences. Lastly, the treatment distribution of the excluded values (or even if ceiling values are entered instead, as suggested above) should be described. I did not see a response to this aspect of my question in the authors’ response.

From the response to Reviewer #1

Line 164 to 165: Why were they excluded?

>>> Trials were ended immediately if latencies exceeded 30s. Although there is always

a cost in applying cut-off thresholds, 30s is often used in studies where calves have to

make a choice (e.g. Lecorps et al., 2018). We subsequently felt exclusion of ≥30s

latencies was the most appropriate approach. Whilst this reduced our data size slightly

and prevented us from capturing extremely aversive calf responses, including ≥30s

latencies may have had a disproportionate effect in our models. We therefore chose to

exclude these latencies from the model. To this effect, the section now reads (line 188-

192):

“If the latency for a given trial exceeded 30s, the trial was ended and the calf returned

to the lobby area, to commence the next trial, consistent with previous studies [27].

Latencies exceeding 30s were excluded from subsequent statistical analysis (21/588

trials excluded). Whilst cut-off thresholds prevent the capture of extremely aversive calf

responses, the exclusion of these latencies ensured our models were not

disproportionately influenced by extreme values.

Line 171: Why are these excluded?

We excluded calves that took greater than 60s to drink for a similar reason – given the

rate of consumptions noted previously, we considered 60s to consume the reward to

be a sufficiently long time to allow for consumption. A cut-off time was necessary to

maintain experimental timings, particularly when multiple calves per session needed to

be tested sequentially. Again, data points ≥60s drinking duration were excluded to

eliminate the effects of extreme values in our models, therefore taking a conservative

approach.

Response to Reviewer #3:

Why were the calves/trials where milk drinking persisted for longer than one minute

excluded?

o What was the number of calves effected by each exclusion of data? How did those

relate to treatment (e.g., Ln 165, 171)?

>>> Trials were ended immediately if latencies exceeded 30s. Although there is always

a cost in applying cut-off thresholds, 30s is often used in studies where calves have to

make a choice (e.g. Lecorps et al., 2018). We subsequently felt exclusion of ≥30s

latencies was the most appropriate approach. Whilst this reduced our data size slightly

and prevented us from capturing extremely aversive calf responses, including ≥30s

latencies may have had a disproportionate effect in our models, making detection of

differences between groups difficult. We therefore chose to exclude these latencies

from the model. To this effect, the section now reads (line 187-192):

“If the latency for a given trial exceeded 30s, the trial was ended and the calf returned

to the lobby area, to commence the next trial, consistent with previous studies [27].

Latencies exceeding 30s were excluded from subsequent statistical analysis (21/588

trials excluded). Whilst cut-off thresholds prevent the capture of extremely aversive calf

responses, the exclusion these latencies ensured our models were not

disproportionately influences by extreme values.”

7. PLOS authors have the option to publish the peer review history of their article (what does this mean?). If published, this will include your full peer review and any attached files.

Reviewer #1: No

Reviewer #3: No

---

## [Author Response · Author response to Decision Letter 2]

13 Jun 2024

Reviewer #1: Thanks for your responses to the comments. I understand in science, it is sometimes necessary to "push" the boundaries; however, the applicability of this is limited. This should be discussed especially when producers aren't even doing "easy" methods of pain control (i.e. local + NSAID)

>>> We agree that the current practicalities of applying this research are difficult given the context. We have added the following to the end of the discussion to reflect this. 

“With the current low levels of NSAID use on farms during disbudding procedures [17], additional research is also required before cold therapy can be practically applied on farms.”

Line 19: Please add in the duration of the contact time

>>> We are unable to provide an exact timing here as the timing varies between each calf and session. 

Line 95 to 97: I find this revised sentence confusing. I would delete that other studies have used 15 calves previously as it doesn't add meaning

>>> We agree and we have therefore decided to remove the sentence completely. 

 

Reviewer #3: Thanks to the authors for including the new figure, refining the use and predictions associated with the approach-aversion methodology and clarifying the conclusions about the new information generated about cold therapy. I have one outstanding concern in the R2 version.

I find the approach to excluding data (latency exceeded 30s, drinking exceeded 60s) concerning, as written. I’ve pasted the responses to Reviewers 1 and 3 about this topic below. From what I understand thus far, I am concerned about the approach taken, to exclude these data, has on the conclusions drawn.

A priori or post-hoc decision. Firstly, for the trials ended if latency was greater than 30s and drinking events were capped at 60s, I think it is important to know if this decision was made a priori or post hoc. As the authors point out, these kinds of cutoffs are common when running trial-based experiments. However, if these data were actually collected, then excluding them and imposing a 30s or 60s end was a post-hoc choice, this is worrisome to me. From my perspective, trimming a dataset to remove extreme values post hoc is not best practice.

>>> Apologies for the lack of clarity here. In both cases, this was an a priori decision.

For latency to enter the pen:

We apologise that we didn’t make this clearer, we did not collect values past cutoffs. Limiting latency to 30s cutoff was necessary given the time constraints we had during the experiment. As a result, we do not have actual extreme values, and 30 seconds represented a ceiling value. 

For drinking duration:

We apologise for the lack of clarity. A trial was ended when a calf stopped drinking before finishing the milk reward, and did not recommence drinking within 60 seconds. This is for a similar reason to the 30s latency above and was also a priori - we only recorded the ceiling value of 60s in this case.

To emphasise the a priori nature of these decisions, we have amended the data collection section (lines 180-183, 189-191). It now reads: 

“We took the a priori decision that, if a calf took more than 30s to enter the pen and contact the bottle, this was interpreted as a ‘no-go’ response, ending the trial. The calf would then be returned to the lobby area to commence the next trial, consistent with previous studies [27].”

and:

“If the calf stopped drinking before the milk bucket was empty, calves were given one minute to reinitiate drinking, otherwise the trial ended, with a 60 second ceiling value recorded (27/594 trials, 14/27 of which were ambient trials)”

Alternatives to exclusion. I appreciate that extreme values can affect statistical models, but exclusion should be a last resort. It is entirely possible that these responses represent an important part of the biological response of the calf. As the authors note, these represent “extremely aversive calf responses” and this was part of the question being asked. As such, exclusion is not “taking a conservative approach” as the authors describe it, at least from my perspective it is not conservative. To me, there are several options that would be preferable to the current approach of complete exclusion from a given model. Even when thresholds are set a priori, I prefer to enter a ceiling value for animals that do not meet or exceed a predetermined threshold, for example 31s for latency and 61s for drinking events, in this case (1 s beyond the cutoff). This approach allows the ceiling effect to be included while still meeting some of the assumptions of the statistical tests (e.g. normality). Another option, especially if data were collected beyond the thresholds (that is the decision about thresholds were post hoc) and actual ceilings were reached (e.g. say 80s drinking event), would be to keep all values in the dataset and explore data transformation in parametric models or use of non-parametric tests to address the model fit concerns the authors describe. Lastly, another approach I admire is to present the model outcomes with and without outliers (or extreme values in this case) included. This allows the reader to see their effect for themselves and make their own decision about how much they trust the findings.

>>> We thank the reviewer for these suggestions. As mentioned above, we do not have data collected beyond the thresholds due to the a priori nature of these decisions. Trials which ended due to these cut-offs do not have corresponding values for the ‘post-drinking contact duration’ or ‘nipple contacts’ metric, hence the ‘NAN’ values in the supplementary raw data file. To make that clearer we have added (lines 197-199):

“The number of nipple contacts or post-drinking contact duration were not recorded for trials which ended due to latency exceeding 30s or pauses in milk reward consumption exceeding 60s.” 

In light of the reviewers’ concerns, we have amended our statistical analysis to include 30 second ceiling values (n=15) in the latency model and 60s ceiling values for drinking duration (n=27).

- For drinking duration, the inclusion of the 60s ceiling values had very little influence on post-disbudding statistics but was significant prior to disbudding. The t and p values have been amended accordingly to reflect the new model (lines 260-266).The pre-disbudding section now reads:

“Prior to disbudding, treatment pack had no influence on latency (t = -1.352, P = 0.187), post-drinking contact duration (t = -1.745, P = 0.086) or nipple contacts (t = -0.117, P = 0.907), but did influence drinking duration (estimate ± SE = 0.09 ± 0.05 s, t = 2.048, P = 0.041), suggesting calves took longer to finish drinking the milk reward with the cool pack prior to disbudding..”

We also reran the permutation statistic regarding the emptying rates of the buckets on each side of the area. With the inclusion of the 60s ceiling values (n=27), there is no longer a statistically significant difference between the bucket emptying rates. This section has now been edited to reflect this (lines 292-294):

“On average, the two buckets were not found to empty at different rates (permutation test, P = 0.088), suggesting that the emptying rates of the two buckets does not help to explain the differences in colour.”

- For Latency, regarding the 30s ceiling values for the Latency post-disbudding model, we particularly like the suggestion of providing statistical metrics for both models, with and without these 30s ceiling values. We have amended our methods such that the model that includes the 30s ceiling values is now the default model 

(with p and t values amended to reflect this – line 272). We have also removed the sentences relating to the interaction term (p=0.071) in the post disbudding results section and discussion section, as the new model does not produce such p values of note, and replaced it with the following results (lines 279-281):

“Furthermore, no significant interaction effect between milk volume and treatment was observed for latency (P = 0.573), drinking duration (P = 0.713), post-drinking contact duration (P = 0.846) or nipple contacts (P = 0.831).” 

The second model that excludes these a priori ceiling values (as in the original manuscript) has been included as supplementary material as it improved the model fit, allowing readers to draw their own conclusions regarding the two models. This second model has been referred to in the methods section on (lines 238-241):

“For latency, a second post-disbudding GLMM was also constructed, which excluded the 30 second ceiling values due to concerns about the their influence on model fit and interpretability. This model better fit the latency data with regards to AIC/BIC criteria and Q-Q plots (see supplementary table (S4)).”

and in the results section on (lines 273-275):

“The influence of pack temperature on changes in latency was still observed when 30 second ceiling values were excluded from the analysis (t = -2.11, 95% CI: -0.181, -0.007, P = 0.035; Fig. S4).”

The raw data file has also been amended to include the 30s/60s ceiling values, and will be reuploaded on resubmission.

Treatment differences. Lastly, the treatment distribution of the excluded values (or even if ceiling values are entered instead, as suggested above) should be described. I did not see a response to this aspect of my question in the authors’ response.

>>> We apologise for missing this aspect of your previous question. We have added a breakdown of the treatment differences in the data collection section (lines 184-185)

For latency: “...for subsequent statistical analysis (15/594 trials, 7/15 of which were ambient trials).”

For drinking duration: “...with a 60 second ceiling value recorded (27/594 trials, 14/27 of which were ambient trials”. 

We should also add, the number of latency trials recorded as 30s was recorded erroneously as 37 in the previous version (as it included trials for testing/habituation phase which were never meant for inclusion in the statistical analysis). It is now correctly displayed as 15 trials. A further 6/594 trials had latencies recorded as ‘nan’due to technical recording difficulties during the trial, as so were excluded. Previously, ratios as above were given as out of 588 (to account for the 6 excluded trials), but we have made these 6 exclusions more explicit, giving ratios out of 594 instead and including a comment about the 6 excluded trials. This additional detail has been added to the results on (lines 269-270): 

“Six trials were excluded following data collection due to technical issues, resulting in a total of 588 trials across all individuals.”

We would also like to note that we have amended slightly the plots in figure 3, as the previous submission had an error (the title for plot B had ‘A)’ in the title by mistake. This has been amended and reuploaded. The uploaded plot for Latency shows the graph with 30s ceiling latencies included.

---

## [Decision Letter · Decision Letter 3]

26 Jun 2024

Cold therapy and pain relief after hot-iron disbudding in dairy calves

PONE-D-23-37809R3

Dear Dr. Lecorps,

We’re pleased to inform you that your manuscript has been judged scientifically suitable for publication and will be formally accepted for publication once it meets all outstanding technical requirements.

Kind regards,

Angel Abuelo, DVM, MRes, MSc, PhD, DABVP (Dairy), DECBHM

Academic Editor

PLOS ONE

Additional Editor Comments (optional):

Reviewers' comments:

Reviewer's Responses to Questions

**Comments to the Author**

1. If the authors have adequately addressed your comments raised in a previous round of review and you feel that this manuscript is now acceptable for publication, you may indicate that here to bypass the “Comments to the Author” section, enter your conflict of interest statement in the “Confidential to Editor” section, and submit your "Accept" recommendation.

Reviewer #1: (No Response)

Reviewer #3: All comments have been addressed

2. Is the manuscript technically sound, and do the data support the conclusions?

Reviewer #1: Partly

Reviewer #3: Yes

3. Has the statistical analysis been performed appropriately and rigorously? 

Reviewer #1: Yes

Reviewer #3: Yes

4. Have the authors made all data underlying the findings in their manuscript fully available?

Reviewer #1: Yes

Reviewer #3: Yes

5. Is the manuscript presented in an intelligible fashion and written in standard English?

Reviewer #1: Yes

Reviewer #3: Yes

6. Review Comments to the Author

Reviewer #1: Thanks for your responses. I would still suggest adding in the exact timing even if there is a range into the abstract.

Reviewer #3: (No Response)

7. PLOS authors have the option to publish the peer review history of their article (what does this mean?). If published, this will include your full peer review and any attached files.

Reviewer #1: No

Reviewer #3: No

---

## [Editor Report · Acceptance letter]

3 Jul 2024

PONE-D-23-37809R3 

PLOS ONE

Dear Dr. Lecorps, 

I'm pleased to inform you that your manuscript has been deemed suitable for publication in PLOS ONE. Congratulations! Your manuscript is now being handed over to our production team.

Kind regards, 

on behalf of

Dr. Angel Abuelo 

Academic Editor

PLOS ONE